# Terazosin Interferes with Quorum Sensing and Type Three Secretion System and Diminishes the Bacterial Espionage to Mitigate the *Salmonella* Typhimurium Pathogenesis

**DOI:** 10.3390/antibiotics11040465

**Published:** 2022-03-30

**Authors:** Wael A. H. Hegazy, Ibrahim M. Salem, Hadil Faris Alotaibi, El-Sayed Khafagy, Doaa Ibrahim

**Affiliations:** 1Department of Microbiology and Immunology, Faculty of Pharmacy, Zagazig University, Zagazig 44519, Egypt; 2Department of Medicinal Chemistry, Faculty of Pharmacy, Suez Canal University, Ismailia 41522, Egypt; dr_ibrahim_m@yahoo.com; 3Department of Pharmaceutical Sciences, College of Pharmacy, Princess Nourah Bint AbdulRahman University, Riyadh 11671, Saudi Arabia; hfalotaibi@pnu.edu.sa; 4Department of Pharmaceutics, College of Pharmacy, Prince Sattam Bin Abdulaziz University, Al-kharj 11942, Saudi Arabia; e.khafagy@psau.edu.sa; 5Department of Pharmaceutics and Industrial Pharmacy, Faculty of Pharmacy, Suez Canal University, Ismailia 41522, Egypt; 6Department of Nutrition and Clinical Nutrition, Faculty of Veterinary Medicine, Zagazig University, Zagazig 44511, Egypt; doibrahim@vet.zu.edu.eg

**Keywords:** terazosin, *Salmonella* Typhimurium, bacterial espionage, anti-virulence, quorum sensing

## Abstract

*Salmonella enterica* is an invasive intracellular pathogen and hires diverse systems to manipulate its survival in the host cells. *Salmonella* could eavesdrop on the host cells, sensing and responding to the produced adrenergic hormones and other neurotransmitters, which results in the augmentation of its virulence and establishes its accommodation in host cells. The current study aims to assess the anti-virulence effect of α-adrenergic antagonist terazosin on *S.* Typhimurium. Our findings show that terazosin significantly reduced *S.* Typhimurium adhesion and biofilm formation. Furthermore, terazosin significantly decreased invasion and intracellular replication of *S.* Typhimurium. Interestingly, in vivo, terazosin protected the mice from *S.* Typhimurium pathogenesis. To understand the terazosin anti-virulence activity, its effect on quorum sensing (QS), bacterial espionage, and type three secretion system (T3SS) was studied. Strikingly, terazosin competed on the membranal sensors that sense adrenergic hormones and down-regulated their encoding genes, which indicates the ability of terazosin to diminish the bacterial eavesdropping on the host cells. Moreover, terazosin significantly reduced the *Chromobacterium violaceum* QS-controlled pigment production and interfered with the QS receptor Lux-homolog *Salmonella* SdiA, which indicates the possible terazosin-mediated anti-QS activity. Furthermore, terazosin down-regulated the expression of T3SS encoding genes. In conclusion, terazosin may mitigate *S.* Typhimurium virulence owing to its hindering QS and down-regulating T3SS encoding genes besides its inhibition of bacterial espionage.

## 1. Introduction

Since the great achievement of the discovery of the first antibiotic, the burden of bacterial infections has been greatly reduced [1]. However, the development of resistance to almost all known classes of antibiotics diminishes the significance of this great achievement [2]. This mandates the innovation of new approaches to overcome the increasing bacterial resistance. Mitigating bacterial virulence is one of these interesting approaches, and its efficacy has been elucidated [3,4,5,6,7,8]. Bacterial cells employ a diverse array of virulence factors, such as bacterial structures, production of an arsenal of extracellular enzymes, and biofilm formation [8,9,10]. Bacteria communicate with each other using inducer/receptor chemical signaling systems to orchestrate their virulence factors. These systems are known as quorum sensing (QS) [11,12,13,14]. Both Gram-negative and Gram-positive bacteria hire QS systems to control motility, biofilm formation, production of enzymes, and other virulence factors [9,15]. Generally, Gram-negative often use diverse autoinducers (AIs), mainly acyl-homoserine lactones (AHLs), which can be detected by QS receptors, mainly LuxR-type receptors [9,16,17]. LuxR-AIs complexes bind to short DNA sequences called lux boxes, regulating the expression of virulence genes [16,18].

*Salmonella enterica* belongs to the *Enterobacteriaceae* family and is a facultative anaerobic intracellular Gram-negative bacterium. *S. enterica* pathogenesis is diverse, from local gastroenteritis caused by *S. enterica* serovars Typhimurium and Enteritidis to serious systemic typhoid fever, caused by *S. enterica* serovar Typhi or Paratyphi [19,20,21]. *S. enterica* virulence factor encoding genes are clustered in particular positions on the bacterial chromosome called *Salmonella* Pathogenicity Islands (SPIs) [19,22]. *S. enterica* employs an intricate type III secretion system (T3SS) to mediate distinct functions through its course of pathogenesis. However, there are several SPIs, but two main SPIs encode two types of T3SS to translocate various effectors in different stages of pathogenesis [19,23]. SPI1-T3SS translocates SPI1 effectors to establish the *Salmonella* invasion, followed by SPI2-T3SS, which translocates the SPI2 effectors to facilitate the *Salmonella* intracellular survival inside host cell phagosomes [20,23,24]. Furthermore, to control its pathogenesis, *S. enterica* uses QS, which regulates biofilm formation, motility, and other virulence factors [14,25]. However, *S. enterica* does not encode an AHL synthase but employs the LuxR homolog SdiA to detect several AHLs with various acyl chain lengths and C-3 substituents [17,25,26,27]. Importantly, SdiA can sense the AHLs produced by other bacterial species, playing a crucial role in QS-controlled *Salmonella* pathogenesis [17,26].

An observed cross talk between bacterial and host cells determines the infection outcome. Bacterial membrane sensors sense the neighboring chemical changes facilitating bacterial accommodation inside the host cells [28,29,30]. Interestingly, AIs that are sensed by QS receptors to regulate bacterial virulence also cross-talk with the adrenergic hormones activating the same signaling pathway in the host cells [29,31]. There is cumulative evidence that the Gram-negative bacteria sense the host neuroendocrine stress hormones, enhancing the bacterial virulence [28,29,32]. In other words, AIs synthesized by bacteria induce the neuroendocrine hormone production in host cells, and in turn, bacteria can sense neuroendocrine hormones augmenting bacterial virulence. In this context, blocking the adrenergic receptor could diminish the bacterial-receptor-based sensing and as a result, mitigate the bacterial virulence [29,32]. So, it has been hypothesized that adrenergic blockers could be candidates to diminish the bacterial espionage on host cells and hence mitigate the bacterial virulence [33,34].

*Salmonella* arouse all these interplayed mechanisms using T3SS, QS systems, and bacterial espionage on host cells to defeat their immunity, causing more aggressive infections [17,22,25]. In this direction, the approach curtailing bacterial virulence employing anti-virulence agents at lower concentrations than their minimum inhibitory concentrations (sub-MIC) offers several advantages [8,35,36]. First, this approach does not force the bacteria to develop resistance as it does not affect bacterial growth [37,38,39,40,41,42,43]. Second, the bacterial normal flora is not destroyed [9,44]. Third, it gives the chance to the immunity to eradicate the virulence-mitigated bacteria [45,46,47,48]. Furthermore, the repurposing of approved drugs as anti-virulent drugs offers additional advantages, including time and cost saving [36,49,50]. Based on the above, it has been proposed that inhibiting the adrenergic receptors, especially α-blockers such as terazosin, could inhibit bacterial receptor-based sensing and diminish bacterial virulence [28,32,33,51]. In a previous study, terazosin significantly mitigated the *Pseudomonas aeruginosa* pathogenesis, which was achieved by interfering with Pseudomonal QS systems [33]. In the current study, the aim was to ensure terazosin anti-virulence activity against *Salmonella*, taking into consideration that *Salmonella* virulence, QS, secretion system, and invasion style are quite different than *Pseudomonas*. Moreover, terazosin effect on *Salmonella* T3SS was evaluated. Most importantly, terazosin effect on *Salmonella* pathogenesis was assessed in comparison to norepinephrine (NE) effect in vitro and in vivo. An in silico molecular study was conducted to evaluate the ability of terazosin to compete with NE on the membranal adrenergic sensors QseC and E on the bacterial surfaces. In the current study, the aim was to show the terazosin ability to interfere with different QS systems in different microbes and, importantly, diminish bacterial espionage, which may be effective in the treatment of gut microbes.

## 2. Results

### 2.1. Detection of the Minimum Inhibitory Concentration (MIC) of Terazosin against S. Typhimurium

The terazosin MIC was detected using the agar dilution method, and the lowest terazosin concentration that inhibited *S.* Typhimurium growth was 4 mg/mL. To exclude the terazosin effect on the *S.* Typhimurium growth, it was grown in its sub-MIC (1/2 MIC). Moreover, the effects of NE and terazosin at sub-MICs on bacterial growth were assessed. The bacterial growth turbidities of treated or untreated *S.* Typhimurium with terazosin at the sub-MIC or NE were determined at 600 nm after 24 h of growth at 37 °C. There was no significant difference between the growths of untreated *S.* Typhimurium and the treated *S.* Typhimurium with terazosin at the sub-MIC or NE (Figure 1). It is worthy to mention that all the following investigations were performed using terazosin at the sub-MIC. 

### 2.2. Terazosin Diminishes S. Typhimurium Adhesion and Biofilm Production

The crystal violet assay was employed to stain the adhered and biofilm-forming bacterial cells after 1 h and 24 h, respectively, in the presence or absence of NE or terazosin at the sub-MIC, and microscope images were taken for the formed biofilms. NE significantly enhanced the *S*. Typhimurium adhesion as well as biofilm formation in comparison to untreated *S*. Typhimurium. However, terazosin at the sub-MIC significantly reduced *S*. Typhimurium adhesion and biofilm formation (Figure 2A,B). Furthermore, microscopic images showed a markedly thicker *S*. Typhimurium biofilms in untreated samples or in the presence of NE, in contrast to the scattered thinner biofilm formed by *S*. Typhimurium treated with terazosin (Figure 2C).

### 2.3. Terazosin Reduces S. Typhimurium Invasion and Intracellular Replication

*S*. Typhimurium internalization within HeLa cells or macrophage was evaluated using the gentamicin protection assay. For invasion assays, HeLa cells were washed and lysed after 1 h infection and the lysates were serially diluted and plated onto MH plates (Figure 3). NE significantly increased the *S.* Typhimurium invasion of HeLa cells. However, terazosin at the sub-MIC significantly diminished the bacterial invasion. For visualization of the terazosin effect on diminishing *Salmonella* invasion, *Salmonella* cells were immunostained with *Salmonella*-O antibodies and secondary cy3 antibodies. The images of the red immunostained *Salmonella* and the bright-field images of HeLa cells were taken and merged (Figure 3A). Obviously, the numbers of the invasive *Salmonella* into HeLa cells were decreased upon treatment with terazosin at the sub-MIC.

Furthermore, *S.* Typhimurium-infected macrophages were washed and then lysed at 2 and 16 h post infection and the lysates were serially diluted and plated onto MH plates. The intracellular bacterial replication was calculated as x-fold (16 h against 2 h) (Figure 4).

### 2.4. Terazosin In Vivo Mitigates the S. Typhimurium Pathogenesis

The terazosin effect at the sub-MIC on *S.* Typhimurium virulence was evaluated by using a mouse infection model. All mice in the negative control groups survived, but only 4 mice out of 10 survived in the mouse group injected with untreated *S.* Typhimurium. Similarly, in the mouse group injected with *S.* Typhimurium treated with NE (50 μg/mL), only 4 mice survived. Terazosin protected 8 mice from *S.* Typhimurium pathogenesis and death. The survival of the mice was plotted by the Kaplan–Meier method, and a Log-rank test was employed to examine the statistical significance. The findings show that terazosin significantly decreased the *S.* Typhimurium capacity to kill mice (*p* = 0.0015) (Figure 5).

### 2.5. Terazosin Anti-Virulence Activity

To evaluate the effect of terazosin on *S.* Typhimurium virulence, several approaches have been proposed, including its effect on QS, T3SS, and/or bacterial espionage to adrenergic hormones.

#### 2.5.1. Anti-QS Activity

##### Terazosin Reduces the Production of QS-Controlled *C. violaceum* Pigment in Comparison to Norepinephrine

The biosensor *C. violaceum* CV026 is regularly used to assess QS activities because it releases the violacein dye in the presence of AHL under the CVi/R QS system control [6,52]. Terazosin (at the sub-MIC) effect on the growth of *C. violaceum* was assessed to exclude any effect on bacterial growth. Moreover, NE effect on *C. violaceum* growth was evaluated. There was no significant difference between untreated *C. violaceum* and NE- or terazosin-treated bacteria (Figure 6A,B). Furthermore, the effects of terazosin at the sub-MIC and NE on violacein production were estimated. NE significantly increased the production of QS-controlled pigment. However, terazosin significantly decreased the pigment release in comparison to untreated *C. violaceum* (Figure 6C).

To attest the anti-QS activities of terazosin, a molecular docking study was conducted to evaluate the binding affinity of terazosin to *C. violaceum* QS CviR. The binding site CviR *C. violaceum* (PDB: 3QP5) demoed lining residues of Gln68, Ile69, Gln70, Tyr80, Tyr88, Ala94, Gln95, Asp97, Arg101, Ile102, Trp111, Arg114, Thr140, Ile153, and Ser155 (Figure 6D). Interestingly, the energy-minimized pose of terazosin accommodates the active site of the CviR *C. violaceum* and elucidates that its binding energy (S = −9.0871 Kcal/mol) exceeds that of the homoacyl lactone (HLC) reference drug (S = −8.3341 Kcal/mol), with a clear indication of the importance of the HBD effect of the 4-amino group of terazosin with the key residue Ile69 beside the importance of the ᴨ–H interaction of the pyrimidine ring with the same residue (Figure 6E).

##### Terazosin Effect on the LuxR Homolog SdiA

*S.* Typhimurium employs the LuxR homolog SdiA to detect several AHLs to modulate the QS-controlled virulence [17,27]. To assess the terazosin anti-QS activities, its effect was evaluated at the sub-MIC on the *sdiA* gene expression in comparison to the effect of NE (Figure 7A). Terazosin (at the sub-MIC) significantly down-regulated the *sdiA* gene expression in comparison to untreated *S.* Typhimurium. Meanwhile, NE significantly up-regulated the expression of *sdiA* gene.

Furthermore, an in silico study was performed to evaluate the terazosin-hindering ability to SdiA. The liner residues of the SdiA *E. coli* (PDB: 4LFU) binding site include Ser43, Cys45, Val57, Phe59, Tyr63, Trp67, Val68, Tyr70, Tyr71, Gln72, Leu77, Asp80, Leu106, Val82, Leu83, Ala109, Ala110, His113, and Leu115 (Figure 7B). Besides, the approximate binding potential scoring function of terazosin (S = −7.0800 Kcal/mol) on the SdiA *E. coli* active site was also comparable with that of the HLC ligand (S = −7.1028 Kcal/mol), which may be attributed to the evenness in the importance of both arene–arene interaction formed between the quinazoline ring of terazosin and Tyr71 residue and the arene–H interaction caused by the HLC ligand with Ala110 residue (Figure 7C).

#### 2.5.2. Terazosin Diminishes *S.* Typhimurium Espionage

*S.* Typhimurium eavesdrops on the host cells using sensor kinases to sense and responds to adrenergic hormones, in particular QseC and QseE sensor kinase, which results in the augmentation of bacterial virulence [29,30]. In this context, the effects of terazosin at the sub-MIC and NE on the expression of both *qseC* and *qseE* genes were assessed (Figure 8A). However, NE significantly increased the expression of both *qseC* and *qseE* genes and terazosin significantly down-regulated their expression.

Furthermore, an in silico docking of both terazosin and NE into the QseC was conducted. QseC *E. coli* (PDB: 3JZ3) terazosin binding pocket lining residues involve Val290, Leu293, Thr295, Arg298, Leu299, Leu305, Gln349, Ser354, Leu353, Leu355, Arg424, Ile425, and Leu428 (Figure 8B). The prominent binding potential of the adrenergic antagonist terazosin with the QseC *E. coli* binding pocket (S = −8.3621 Kcal/mol) transcended the total binding scoring of the adrenergic agonist epinephrine (S = −5.8122 Kcal/mol), although it demoed only one HBD effect of the 4-amino group with Ser297 residue, as clarified in Figure 8C, which explains the proceeding affinity of terazosin against a QseC *E. coli* active site for further investigations.

#### 2.5.3. Terazosin Down-Regulates the Expression of T3SS Encoding Genes 

*S.* Typhimurium employs two types of T3SS to establish its invasion and to survive inside the bacterial phagosomes [19,24]. As a consequence, interfering with this magnificent injectosome has a considerable impact on the *S.* Typhimurium pathogenesis. The terazosin effect at the sub-MIC on the expression of various T3SS type 1 and 2 encoding genes was evaluated in comparison to untreated or NE-treated *S.* Typhimurium (Figure 9). Terazosin significantly down-regulated the expression of the T3SS-tested encoding genes in contrast to NE, which increased their expression.

## 3. Discussion

*S. enterica* is an intracellular bacterium and causes food-borne pathogenesis ranging from gastroenteritis to systematic enteric fever [19,25]. *S. enterica* comprises various serovars that share in a considerable arsenal of virulence factors [19,20]. Besides the magnificent injectosome T3SS, which enables *Salmonella* invasion and survival inside host cells [24,53], *Salmonella* could eavesdrop on the host cells and respond to different catecholamines, adrenergic hormones, and other neurotransmitters [29,30,32]. Furthermore, *Salmonella* QS orchestrates the virulence by sensing several autoinducers AI on LuxR homologs, such as SdiA [17,25,26,27], which results in the augmentation of *Salmonella* pathogenesis. In addition, *S.* Typhimurium has shown increased resistance to different antibiotics [20,24]. This situation mandates innovating and developing new approaches to conquer the *Salmonella* resistance to antibiotics. Mitigating bacterial virulence has been proven as an efficient approach to diminishing the bacterial virulence without stressing the bacteria to develop a resistance, facilitating bacterial eradication [5,6,25,54]. Bacterial virulence could be targeted by various natural compounds and chemical moieties. However, the benefit is maximized when using approved safe anti-virulence agents [5,6,38,40,55]. In this direction, the current study aimed to evaluate the α-adrenergic antagonist terazosin’s anti-virulence activity against *Salmonella*. Especially, its anti-virulence activity was shown against *Pseudomonas aeruginosa* [33].

*Salmonella* can sense adrenergic hormones epinephrine and norepinephrine employing sensor kinases such as QseC and QseE, which impacts by enhancing the pathogenesis [29,30]. It has been shown that norepinephrine can increase *Salmonella*’s in vitro virulence and in vivo pathogenesis [32,56]. So, it is meaningful to study the anti-virulence activity of the α-adrenergic antagonist terazosin in comparison to that of norepinephrine. The effectiveness of targeting bacterial virulence in diminishing resistance development basically depends on avoiding the stress on bacterial growth, as reviewed [12]. To exclude any effect of terazosin on *Salmonella* growth, terazosin was used at the sub-MIC in all experiments. Despite this, neither terazosin at the sub-MIC nor norepinephrine significantly affected the bacterial growth. Terazosin significantly diminished *Salmonella* adhesion and biofilm formation in contrast to norepinephrine, which significantly increased both. Furthermore, terazosin significantly decreased *Salmonella* invasion into HeLa cells and intracellular replication inside macrophages, contrary to norepinephrine, which significantly enhanced bacterial invasion and intracellular replication. An in vivo model has been used to attest the terazosin anti-virulence activity; terazosin significantly protected mice from *S.* Typhimurium when compared to mice injected with *S.* Typhimurium-treated norepinephrine or untreated *S.* Typhimurium. While these findings are in agreement with cumulative data that showed the significant effect of norepinephrine on *S.* Typhimurium [29,30,56], terazosin anti-virulence activity requires to be explored.

QS orchestrates the virulence of both Gram-negative and Gram-positive bacteria. It controls the production of various exocellular enzymes, bacterial motility, the formation of biofilm, and other virulence factors [9,11,16,36]. Based on this, the targeting of QS was shown as a promising approach to diminish the bacterial virulence [4,5,6,9,42]. Violacein pigment is encoded by the *vio* operon, which is QS regulated in *C. violaceum* [52]. Because of the easy observation and quantification of this *C. violaceum* QS-regulated trait, it was employed extensively to screen the effect on QS [33,34,52]. In this context, the terazosin effect at the sub-MIC on the production of the QS-regulated violacein pigment was analyzed to indicate terazosin’s anti-QS activity. Terazosin significantly decreased the release of violacein, in contrast to norepinephrine, indicating the possible interference of terazosin with bacterial QS. Furthermore, in silico molecular docking showed terazosin’s ability to compete on the *C. violaceum* CviR QS receptor.

QS receptors senses their cognate inducers to initiate the bacterial virulence regulation via down- or up-regulation of involved virulence factor encoding genes [9,11,15,16]. However, *S.* Typhimurium does not produce its own inducers. It employs the LuxR homolog SdiA, which detects diverse inducers. *S.* Typhimurium SdiA senses numerous AHLs activating sdiA-regulated genes (srgs) to control its virulence [17,57,58]. In the current study, terazosin showed a marked ability to compete on SdiA and interfere with *S.* Typhimurium QS. Furthermore, terazosin at the sub-MIC down-regulated the *sdiA* gene expression. These findings conclude the possible anti-QS activity of terazosin that results in diminishing of the *S.* Typhimurium virulence. It is worthy to mention that norepinephrine up-regulated the *sdiA* gene. The role of SdiA in controlling *S.* Typhimurium adhesion and biofilm formation has been previously evaluated [21,25]. In great compliance, terazosin down-regulated *sdiA* gene and diminished the adhesion and biofilm formation of *S.* Typhimurium, in contrast to norepinephrine.

The increasing evidence of bacterial spying on the eukaryotic host cell is an additional mechanism that bacteria employ to establish their invasion [29,30,32]. Gram-negative bacteria recruit sensors on their membranes to sense diverse catecholamines and neurotransmitters. In particular, *E. coli* and *Salmonella* spp. use membrane sensor kinases QseC and QseE to sense adrenergic hormones epinephrine and norepinephrine [29,30]. In the current study, norepinephrine increased the expression of *qseC* and *qseE* genes, which reflects in increasing bacterial virulence. This finding is in agreement with previous studies [29,30,32,59]. However, terazosin significantly decreased the expression of both *qseC* and *qseE* genes. Moreover, terazosin showed a considerable binding affinity to the sensor kinase QseC in comparison to norepinephrine. These results give evidence of terazosin interreference with bacterial spying on the host cells. Additionally, QS inducers interplay with adrenergic hormones to trigger bacterial virulence [31], so blocking the adrenergic receptors inhibits the *S.* Typhimurium-receptor-based sensing and diminishes its virulence [28,29,31]. Terazosin mitigates bacterial virulence as it antagonizes the adrenergic-hormone-induced pathogenesis basically by decreasing their release and then by competing with them on their sensors on bacterial surfaces.

In addition to QS, *S.* Typhimurium recruits complicated systems to regulate its pathogenesis. *Salmonella* spp. employ a sophisticated T3SS injectosome, which is encoded mainly by clusters of genes arranged in two pathogenicity islands (SPIs) [19,53]. The two types of T3SS play distinct functions, as T3SS type 1 plays a role in easing the invasion of *Salmonella* and the second type guarantees survival inside the host immune cells [19,53,60,61]. Moreover, T3SS type 2 is encoded by more than 30 SPI2 genes that are arranged in two operons; the first operon encodes the secretion system apparatus (Ssa) components, such as *ssaE* and *ssaJ*, and also includes transcriptional unit expressing secretion system effectors (Sse), such as *sseI*, *sseF*, and *sseJ* genes, and secretion system chaperones (Ssc), such as *sscA* gene. Another operon encodes the regulatory secretion system SsrAB [19,60,62]. In the present study, terazosin at the sub-MIC significantly reduced the *Salmonella* intracellular replication inside the macrophages, which could be due to interference with SPI2 gene functions. Furthermore, these results were confirmed using a qRT-PCR, which revealed the significant effect of terazosin on the down-regulation of SPI2 genes *ssrB, ssaE*, *sseF*, *ssaJ*, *sseI*, *sseF*, *sseJ*, and *sscA*. The roles of all these genes in *Salmonella* pathogenesis have been well characterized [19,24,62]. After *Salmonella* enters the host cells, it forms Salmonella-induced filaments (SIFs) to reach the host endosome, establishing its intracellular replication [63]. The biogenesis of SIFs depends on the activity of SPI2 effector proteins that are secreted through T3SS type 2 as *siF* genes encode SIF formatting proteins SifA and SifB [61,63,64]. Importantly, current observations show that terazosin significantly decreased the expression of *sseF*, *sifA*, and *sifB* in *Salmonella*, which could explain the lowered intracellular survival of *Salmonella* within macrophages. Strikingly, norepinephrine significantly increased the expression of all the testes genes, which could also explain the role of norepinephrine in enhancing the pathogenesis of *Salmonella*.

## 4. Materials and Methods

### 4.1. Microbiological Media, Chemicals, and Bacterial Strains 

All the chemicals used were of pharmaceutical grade. Mueller Hinton (MH) agar and broth, Luria–Bertani (LB) agar and broth, Tryptic Soy Agar (TSA), and Tryptone soy broth (TSB) were purchased from Oxoid (Hampshire, UK). Dulbecco’s Modified Eagle’s Medium (DMEM) medium, DL-norepinephrine hydrochloride (CAS Number: 55-27-6), and N-hexanoyl-DL-homoserine lactone AHL (CAS Number: 106983-28-2) were obtained from Sigma-Aldrich (St. Louis, MO, USA). *Salmonella enterica* serovar Typhimurium (NCTC 12023) and *Chromobacterium violaceum* CV026 (ATCC 31532) were used in this work. 

### 4.2. Determination of Terazosin MIC

The agar dilution method was employed to detect the MIC of terazosin against *S.* Typhimurium and *C. violaceum* according to the Clinical Laboratory and Standards Institute Guidelines (CLSI, 2012) as described earlier [20,65]. Briefly, the tested strains were incubated overnight in TSB and then diluted in MH broth to turbidity approximately equivalent to 0.5 McFarland Standard. Bacterial suspensions were diluted 1:10 to prepare standard inoculum (1 × 10^4^ CFU/mL), which was spotted on the MH agar plates provided with different terazosin concentrations. Free MH agar plates without terazosin were kept without bacterial culturing as a negative control or spotted with *Salmonella* as positive control, and the experiment was conducted as triplicate. After overnight incubation at 37 °C, the MIC of terazosin was considered the lowest concentrations that inhibit the bacterial growth. 

### 4.3. Terazosin Effect at the Sub-MIC on Bacterial Growth

To evaluate the terazosin anti-virulence activity, the effect of terazosin on bacterial growth should be excluded. Furthermore, the effect of norepinephrine (NE) at a concentration of 50 μg/mL was evaluated on the growth of tested strains [56]. The effect of terazosin at the sub-MIC or NE on *S.* Typhimurium and *C. violaceum* was evaluated as described previously [4,5,39]. Bacterial suspensions (1 × 10^8^ CFU/mL) of tested strains with turbidity equivalent to 0.5 McFarland Standard were used to inoculate LB broth provided with or without NE (50 μg/mL) or terazosin (sub-MIC). The optical densities of inoculated cultures were measured at 600 nm after 24 h. The experiment was conducted in triplicate, and the data are presented as the mean ± the standard error. To test the statistical significance, a one-way ANOVA test, followed by Tukey’s multiple comparison post-hoc test, was employed, where *p* < 0.05 was considered statistically significant.

### 4.4. Evaluation of Terazosin Effect on S. Typhimurium Adhesion and Biofilm Formation

*S*. Typhimurium overnight cultures were diluted in TSB broth to cell density 1 × 10^6^ CFU/mL (OD600 = 0.4) for adhesion and biofilm formation assay. Then, 100 μL aliquots of *S*. Typhimurium prepared suspensions were transferred to wells of polystyrene microtiter plates filled with 200 μL of LB broth provided with or without NE (50 μg/mL) or terazosin (sub-MIC) in the presence of 0.001 μM AHL [25]. The microtiter plates were kept at 37 °C for 1 h and 24 h to evaluate the bacterial adhesion and biofilm formation, respectively [20,25,42,57]. The incubated plates were washed three times with sterile saline to remove nonadherent bacterial cells, fixed at 60 °C for 30, min and then stained for 20 min with crystal violet (1%). After washing out the excess dye, crystal violet was extracted with ethanol and optical densities were measured at 590 nm. The experiments were conducted in triplicate, and the data are presented as the mean ± the standard error. To test the statistical significance, a one-way ANOVA test, followed by Tukey’s multiple comparison post-hoc test, was employed, where *p* < 0.05 was considered statistically significant.

To visualize the effect of terazosin at the sub-MIC or NE on biofilm formation, the above protocol was employed to allow *S*. Typhimurium to form biofilms on sterile coverslips in the presence or absence of terazosin at the sub-MIC or NE [20,37,54]. The formed biofilms on coverslips were fixed, stained with crystal violet, and examined using light microscope (400× magnification) using a Leica DM750 HD digital microscope (Mannheim, Germany).

### 4.5. Evaluation of Terazosin Effect on S. Typhimurium Invasion and Intracellular Replication 

A gentamicin protection assay was employed to evaluate the *S*. Typhimurium internalization within different cell lines in the presence or absence of NE (50 μg/mL) or terazosin (sub-MIC), as described earlier [25,57,60]. Twenty-four-well polystyrene plates were seeded with HeLa cells or RAW264.7 at cell densities of 5 × 10^5^ and 2 × 10^5^ cells/well for invasion and intracellular replication, respectively. *S*. Typhimurium overnight cultures were incubated for 4 h at 37 °C, and 300 μL of an inoculum master-mix (1 × 10^5^ bacteria/well) with multiplicity of infection (MOI 1) for cell lines was prepared in DMEM and added to the wells. Bacterial infection was performed in the presence of 0.001 μM AHL and the presence or absence of terazosin at sub-MIC or NE (50 μg/mL). After 30 min, the non-internalized bacterial cells were washed out with pre-warmed phosphate buffer saline (PBS). The extracellular adhered bacterial cells were killed by incubation for 1 h in a medium containing 100 μg/mL of gentamicin. HeLa cells were lysed with Triton X-100 (0.1%) for 15 min at 25 °C for evaluation of bacterial invasion. To count the intracellular-invading bacterial cells, the inoculum and the lysates were serially diluted and plated onto MH plates and the invading *Salmonella* (1 h against inoculum) was determined. For an intracellular replication assay, the infected macrophage cells were washed with PBS and lysed with 0.1% TritonX-100 for 15 min in 25 °C at 2 and 16 h post infection. The inoculum and the lysates were serially diluted and plated onto MH plates. The phagocytosed cell numbers/relative untaken cells (2 h against inoculum) and x-fold intracellular replication (16 h against 2 h) were determined. The experiments were conducted in triplicate, and the data are presented as the mean ± the standard error. To test the statistical significance, a one-way ANOVA test, followed by Tukey’s multiple comparison post-hoc test, was employed, where *p* < 0.05 was considered statistically significant.

To visualize the *Salmonella*-infected HeLa or macrophage cells, bacterial cells were immunostained as described earlier [20,57,61]. HeLa cells or macrophages were infected with *S.* Typhimurium in the absence or presence of NE or terazosin. Infected HeLa cells or macrophages were fixed at 1 h or 16 h post infection with paraformaldehyde (2%), left for 25 min, and then washed with PBS. Bovine serum albumin (BSA) (2%) was added at 25 °C as a blocking solution and the mixture left for 1 h. After washing with PBS, *Salmonella* lipopolysaccharide (LPS) and rabbit anti-Salmonella O antigen (Difco, BD; San Joes, CA, USA) were added to fixed cells. The mixture was left for 3 h and then washed with PBS. For staining *Salmonella* inside HeLa cells, anti-rabbit tagged with cy3 secondary antibody (red fluorescent protein) (Abcam; USA) was added and the mixture was left for 1 h. For staining *Salmonella* inside macrophages, anti-rabbit tagged with green fluorescent protein (GFP) secondary antibody (green fluorescent protein) (Abcam; Eugene, OR, USA) was added, the mixture was left for 1 h, and the macrophages were counter-stained with blue fluorescent diamidino-2-phenylindole dye (DAPI) (Thermo Fisher Scientific; Bothell, WA, USA) and left for 1 h. After the washing the macrophages with PBS, microscopic images were captured using a Zeiss LSM780 confocal laser scanning microscope.

### 4.6. Evaluation of the Effect of Terazosin on the Production of QS-Controlled Violacein Pigment

As previously described [3,6], the overnight cultures of *C. violaceum* CV026 were adjusted in LB broth to 1 at 600 nm. The bacterial suspensions (100 μL) were mixed with same volumes of AHL solutions in the absence or presence of NE (50 μg/mL) or terazosin (sub-MIC) in wells of microtiter plates. After overnight incubation at 25 °C, the plates were dried and the produced dye was extracted using dimethylsulfoxide (DMSO) at 30 °C. A DMSO negative control was prepared. The optical densities of the produced pigment were measured at 590 nm. The experiments were conducted in triplicate, and the data are presented as the mean ± the standard error. To test the statistical significance, a one-way ANOVA test, followed by Tukey’s multiple comparison post-hoc test, was employed, where *p* < 0.05 was considered statistically significant.

### 4.7. In Vivo Evaluation of Terazosin Effect on S. Typhimurium Pathogenesis

The in vivo protective assay using mice as the animal model was used to estimate the effect of terazosin at the sub-MIC or NE (50 μg/mL) on *S.* Typhimurium pathogenesis, as described previously [20,25]. The cell densities of the *S.* Typhimurium overnight culture were prepared to about 1 × 10^8^ CFU/mL in LB broth. Five female albino mouse groups (*n* = 10) with similar weights were recruited. There were two negative groups. In one, the mice were kept un-injected and in the other, the mice were intraperitoneally (ip) injected with 100 μL PBS. The other three groups were ip injected with 100 μL untreated *S.* Typhimurium or *S.* Typhimurium treated with NE (50 μg/mL) or terazosin (sub-MIC). The mice were saved at room temperature, with normal feeding and aeration under the same circumstances. The survival of mice in each group was documented daily for 5 days using the Kaplan–Meier method, and the Log-rank test was employed to attest the statistical significance (*p* < 0.05).

### 4.8. Evaluation of Terazosin on the Expression of S. Typhimurium Virulence Genes

To evaluate the effect of terazosin at the sub-MIC or NE (50 μg/mL) on the regulation of the expression of diverse involved *S.* Typhimurium virulence genes, a quantitative real-time PCR was employed [66,67]. The RNA of terazosin- or NE-treated or untreated *S.* Typhimurium after 24 h culturing was extracted by the Gene JET RNA Purification Kit (Thermoscientific, Waltham, MA, USA) according to the manufacturer instructions. The extracted RNA was stored at −80 °C until use. the SensiFAST™ SYBR^®^ Hi-ROX One-Step Kit (Bioline, London, UK) protocol was applied for analysis employing the StepOne Real-Time PCR system (Applied Biosystem, Foster City, CA, USA). The comparative threshold cycle (ΔΔCt) method was used to calculate the relative gene expression. The levels of relative expression of tested genes were evaluated in S. Typhimurium treated or untreated with NE or terazosin in relation to the housekeeping *gyrB* gene. The primers used in this study are listed in Table 1. The experiments were conducted in triplicate, and the data are presented as the mean ± the standard error. To test the statistical significance, a one-way ANOVA test, followed by Tukey’s multiple comparison post-hoc test, or a two-way ANOVA test, followed by Bonferroni post-hoc test, was employed, where *p* < 0.05 was considered statistically significant.

### 4.9. Target Preparation and Ligand Construction for Docking Analysis

The proposed adrenergic antagonist terazosin and the selected reference ligands homoacyl lactone (HLC) and adrenergic agonist epinephrine were constructed within the MOE2014.0901 software package (CCGTM, Montreal, QC, Canada). A molecular docking study was performed to identify the most preferred binding modes of the terazosin ligand inside 3 active sites: SdiA *E. coli* (PDB: 4LFU) escorted by the HLC ligand, QseC *E. coli* (PDB: 3JZ3) comparable to epinephrine, and CviR *C. violaceum* (PDB: 3QP5) also together with HLC. The constructed ligands had diminished energy through a conjugate-gradient approach of 2000 steps until they reached a root-mean-square gradient convergence of 1 × 10^−3^ Kcal/mol/Å^2^ using MMFF94s partial charges and an MMFF94s-modified forcefield. The selected biological targets SdiA *E. coli* (PDB: 4LFU), QseC *E. coli* (PDB: 3JZ3), and CviR *C. violaceum* (PDB: 3QP5) were obtained from the RCSB-Protein Data Bank. Proteins were prepared via 3D protonation after discharge of the atomic backbone and the side chain of the skeleton, besides autocorrection of atom types, partial charges, and bond connectivity. Finally, the MOE loop modeler was used for modeling missing loops within the LuxR-type QscRs PDB files. Specific binding sites of each target were defined by the MOE-Alfa Site Finder geometrical approach while being refined for including the vital residues reported in the present study.

## 5. Conclusions

Terazosin may diminish *Salmonella* virulence owing to its interference with the QS system and bacterial espionage on host cells besides its ability to down-regulate the expression of T3SS encoding genes. It is documented that adrenergic hormones enhance bacterial pathogenesis and as an antagonist, terazosin could diminish adrenergic-hormone-induced bacterial pathogenesis. However, terazosin also may acquire undesirable pharmacological adverse effects, which require further pharmacological and toxicological studies to evaluate terazosin’s clinical use. In the current study, the anti-virulence activity of terazosin has been proven with the proposed mechanisms, keeping its use as an adjuvant to traditional antibiotics or as a pharmacophore to develop new candidates for further investigation. 

## Figures and Tables

**Figure 1 antibiotics-11-00465-f001:**
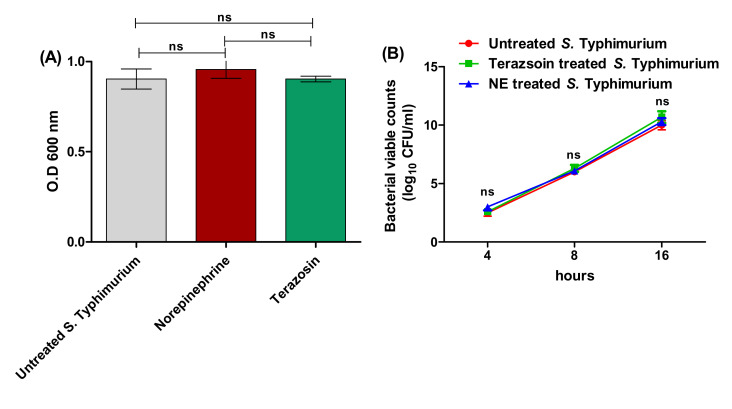
The effect of terazosin at the sub-MIC and norepinephrine on *S.* Typhimurium growth. (**A**) There was no significant difference between the O.D of *S.* Typhimurium growth in the presence of norepinephrine (50 μg/mL) or terazosin (sub-MIC) and the O.D of untreated bacterial growth after 24 h. (**B**) Viable counting of terazosin-treated, NE-treated, or untreated control *S.* Typhimurium cultures at different time periods (after 4 h, 8 h, and 16 h). Neither terazosin nor NE showed a statistical inhibitory effect on *S.* Typhimurium growth. The experiment was repeated in triplicate, and the data are presented as the mean ± the SD. A one-way ANOVA test, followed by Tukey’s multiple comparison post-hoc test, was employed to attest the statistical significance; *p* < 0.05 was considered significant.

**Figure 2 antibiotics-11-00465-f002:**
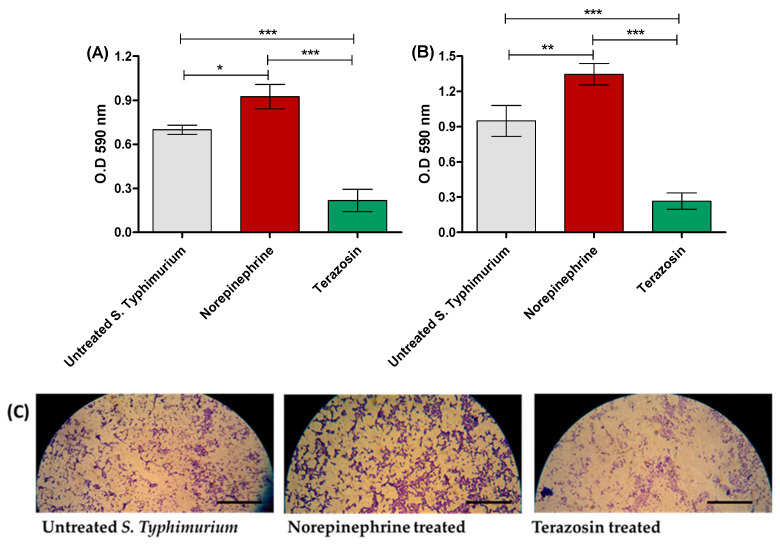
Terazosin diminishes *S.* Typhimurium (**A**) adhesion and (**B**) biofilm formation. Terazosin significantly decreased the bacterial adhesion and biofilm production in comparison to untreated bacteria or norepinephrine-treated bacteria. However, norepinephrine significantly increased bacterial adhesion and biofilm formation. (**C**) Light microscope images of the formed biofilms in the presence of norepinephrine or terazosin. Norepinephrine markedly increased the formed biofilm and terazosin markedly diminished biofilm formation. The experiment was repeated in triplicate, and the data are presented as the mean ± the SD. A one-way ANOVA test, followed by Tukey’s multiple comparison post-hoc test, was employed to attest the statistical significance; *p* < 0.05 was considered significant. *** = *p* < 0.0001; ** = *p* < 0.001; * = *p* < 0.05. Scale bars correspond to 100 μm.

**Figure 3 antibiotics-11-00465-f003:**
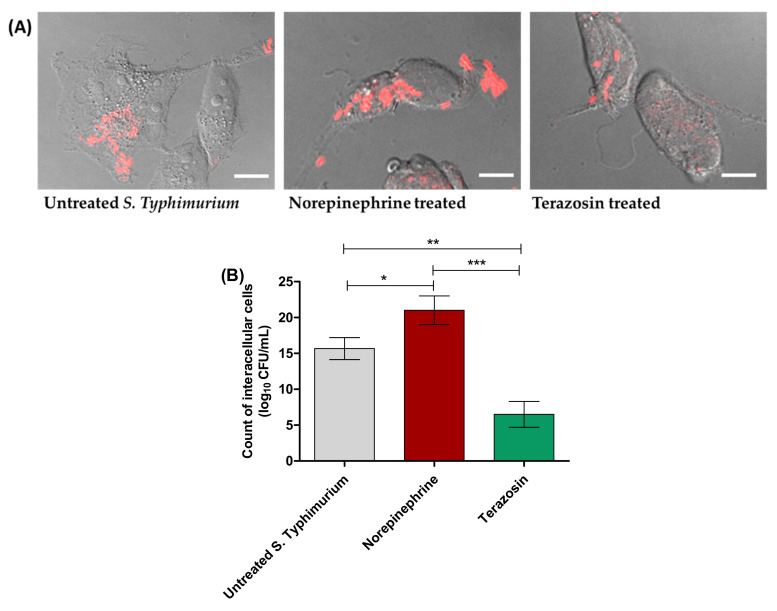
Terazosin diminishes the *S.* Typhimurium invasion into HeLa cells. (**A**) Microscopic images of *S.* Typhimurium in HeLa cells. Norepinephrine markedly increased the number of invading bacterial cells, but terazosin at the sub-MIC diminished the bacterial invasion in comparison to untreated *S.* Typhimurium. (**B**) The HeLa were lysed, and the invading bacterial cells were viably counted. Terazosin significantly reduced the number of invading *S.* Typhimurium, in contrast to norepinephrine, which increased the *S.* Typhimurium invasion. The experiment was repeated in triplicate, and the data are presented as the mean ± the SD. A one-way ANOVA test, followed by Tukey’s multiple comparison post-hoc test, was performed to attest the statistical significance; *p* < 0.05 was considered significant. *** = *p* < 0.0001; ** = *p* < 0.001; * = *p* < 0.05. Scale bars correspond to 10 μm.

**Figure 4 antibiotics-11-00465-f004:**
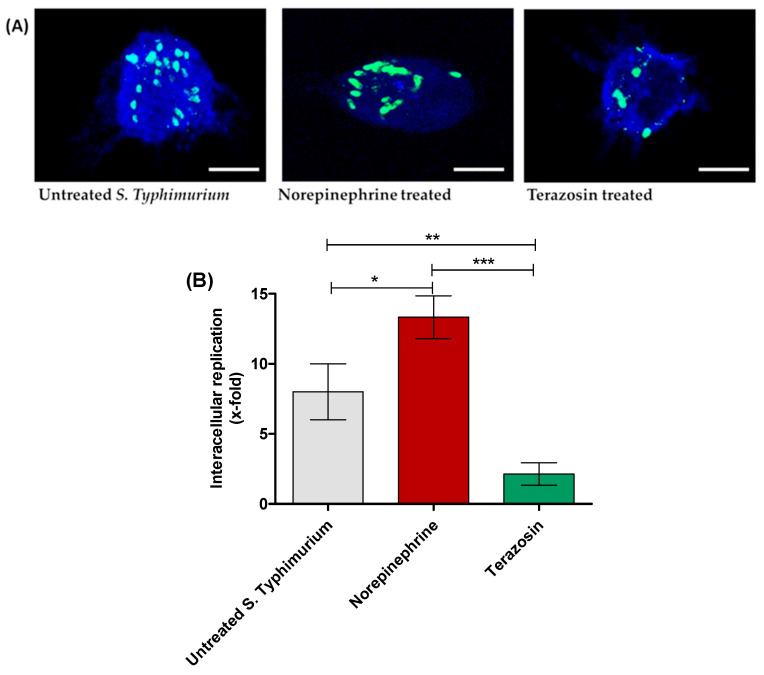
Terazosin diminishes *S.* Typhimurium intracellular replication in raw macrophage cells. (**A**) Microscopic images of *S.* Typhimurium in macrophages. Norepinephrine markedly increased the number of intracellularly replicating bacterial cells and terazosin at the sub-MIC diminished the bacterial intracellular replication in comparison to untreated *S.* Typhimurium. (**B**) The infected macrophages were lysed, and the phagocytosed cell/relative untaken cell percentage and x-fold intracellular replication were calculated. The experiment was repeated in triplicate, and the data are presented as the mean ± the SD. A one-way ANOVA test, followed by Tukey’s multiple comparison post-hoc test, was employed to attest the statistical significance; *p* < 0.05 was considered significant. *** = *p* < 0.0001; ** = *p* < 0.001; * = *p* < 0.05. Scale bars correspond to 100 μm.

**Figure 5 antibiotics-11-00465-f005:**
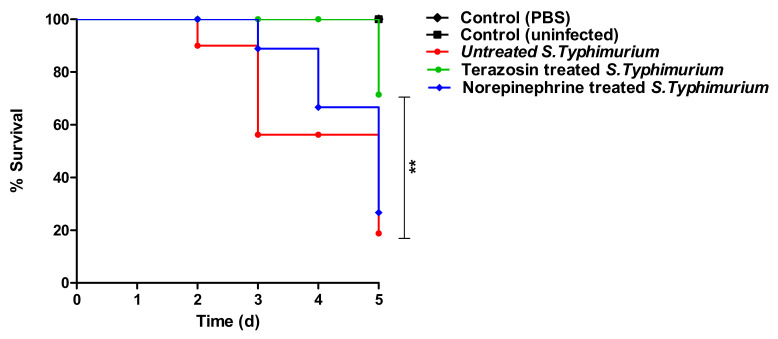
Terazosin protected mice against S. Typhimurium pathogenesis. Mouse groups (*n* = 10) were injected either with 100 μL (2 × 10^6^ CFU/mL) of *S.* Typhimurium treated with norepinephrine (50 μg/mL) or with terazosin at the sub-MIC or untreated *S.* Typhimurium, in addition to the negative control groups of the uninfected mouse group and the mouse group injected with sterile PBS. In mouse groups injected with untreated *S.* Typhimurium or norepinephrine-treated *S.* Typhimurium, only 4 mice survived. All mice survived in the negative control groups. However, terazosin protected 9 mice from *S.* Typhimurium. A Log-rank test for trend *p* = 0.0015 (** = *p* < 0.001).

**Figure 6 antibiotics-11-00465-f006:**
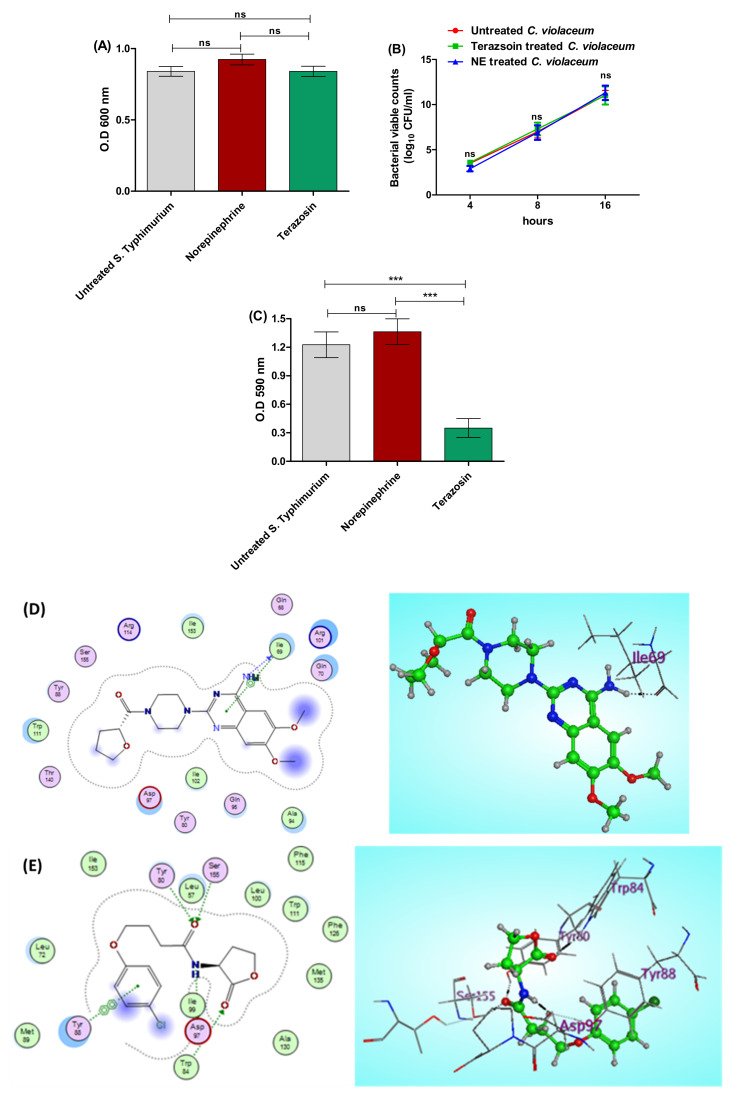
Terazosin diminishes the *C. violaceum* QS encoding violacein. (**A**) Terazosin effect at the sub-MIC and norepinephrine (50 μg/mL) on the growth of *C. violaceum*. (**B**) Viable counting of terazosin-treated, NE-treated, or untreated control *C. violaceum* cultures at different time periods. There was no significant difference between the O.D of viable counting of *C. violaceum* in the presence of norepinephrine or terazosin and that of untreated bacterial growth. (**C**) Violacein pigment production. Terazosin at the sub-MIC significantly decreased the violacein release in comparison to untreated bacteria or norepinephrine-treated bacteria. The experiment was repeated in triplicate, and the data are presented as the mean ± the SD. A one-way ANOVA test, followed by Tukey’s multiple comparison post-hoc test, was employed to attest the statistical significance; *p* < 0.05 was considered significant. *** = *p* < 0.0001. (**D**) 2D and 3D binding modes of terazosin in the CviR *C. violaceum* binding site. (**E**) 2D and 3D binding modes of HLC in the CviR *C. violaceum* binding site. Terazosin showed a marked ability to compete on the *C. violaceum* CviR QS receptor.

**Figure 7 antibiotics-11-00465-f007:**
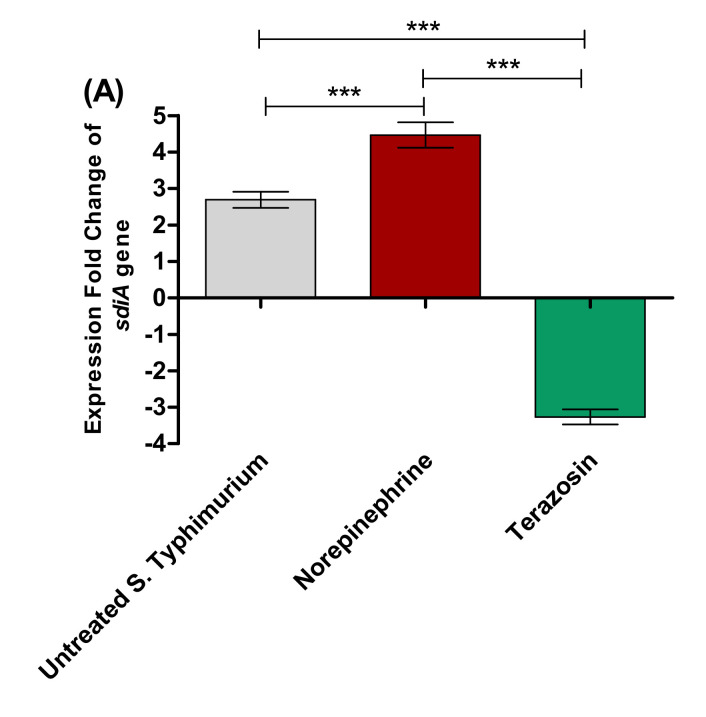
Terazosin interferes with the LuxR homolog QS receptor SdiA. (**A**) Terazosin down–regulated the expression of *sdiA* gene. An Rt-PCR was used to determine the effects of terazosin at the sub–MIC and norepinephrine on the expression of *sdiA* gene. Terazosin significantly decreased the expression of *sdiA* gene in contrast to norepinephrine, which significantly increased its expression. The experiment was repeated in triplicate, and the data are presented as the mean ± the SD. A one-way ANOVA test, followed by Tukey’s multiple comparison post-hoc test, was employed to attest the statistical significance; *p* < 0.05 was considered significant. *** = *p* < 0.0001. (**B**) 2D and 3D binding modes of terazosin in the SdiA *E. coli* binding site. (**C**) 2D and 3D binding modes of HLC in the SdiA *E. coli* binding site. Terazosin showed a marked ability to compete on SdiA, the homolog for LuxR QS, and hinder it.

**Figure 8 antibiotics-11-00465-f008:**
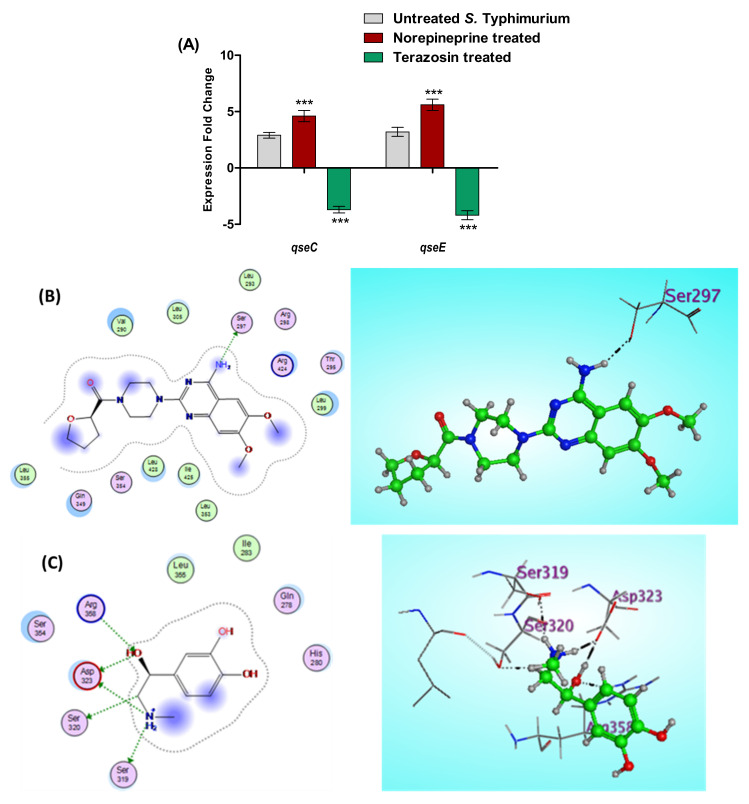
Terazosin interferes with adrenergic hormone sensors. (**A**) Terazosin down–regulated the *qseC* and *qseE* gene expression. An Rt–PCR was used to assess the effect of terazosin at the sub–MIC on the expression of the membranal adrenergic hormone sensor encoding gene. Terazosin significantly decreased the expression of both genes in contrast to norepinephrine, which significantly increased their expressions. The experiment was repeated in triplicate, and the data are presented as the mean ± the SD. A two-way ANOVA test, followed by a Bonferroni post-hoc test, was used to attest the statistical significance; *p* < 0.05 was considered significant. *** = *p* < 0.0001. (**B**) 2D and 3D binding modes of terazosin in the QseC *E. coli* binding site and (**C**) 2D and 3D binding modes of norepinephrine in the QseC *E. coli* binding site. Terazosin showed a higher affinity to bind and compete on the QseC sensor kinase in comparison to norepinephrine.

**Figure 9 antibiotics-11-00465-f009:**
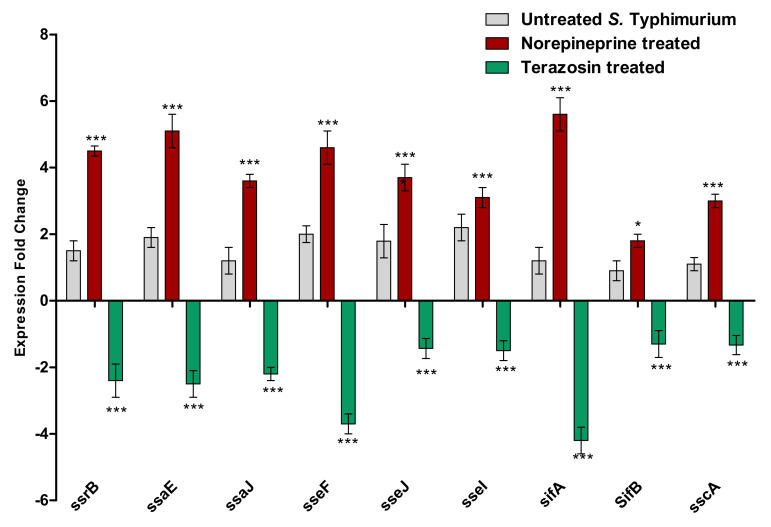
Terazosin down–regulated the T3SS encoding gene expression. Terazosin at the sub–MIC reduced significantly the expression of T3SS encoding genes. However, norepinephrine up–regulated the expression of the tested genes. The experiment was repeated in triplicate, and the data are presented as the mean ± the SD. A two-way ANOVA test, followed by a Bonferroni post-hoc test, was used to attest the statistical significance; *p* < 0.05 was considered significant. *** = *p* < 0.0001; * = *p* < 0.05.

**Table 1 antibiotics-11-00465-t001:** The primers used in this study.

Target Gene	Primer Sequence: 5′-3′	Gene Significance	Reference
** *gyrB* **	F: GTGATCAGCGTCGCCACT R: GCGCGGTGATCAGCGTC	Housekeeping	[20]
** *ssrB* **	F: CGCAGGTGCTAATGGCTATG R: TTTGCAATGCCGCTAACAGA	SPI2-expression regulation	[20]
** *ssaE* **	F: CCGCAGCAATATCAGCAAAA R: AAGTGCGCTGTTATGGTAACGA	SPI2-intracellular replication	[20]
** *ssaJ* **	F: TGTCGAGCAGTCGCAGTTTATTA R: TGCCTATGCGGATAACCGTTA	SPI2-intracellular replication	[20]
** *sseF* **	F: TCAGGAATCGCTATTTCTATG R: GTCAGGCTAACGGAGGTAA	SPI2-intracellular replication	[20]
** *sseJ* **	F: AATAAATCACATCCCAAGC R: ACTCAGTCCAGGTAAATCC	SPI2-intracellular replication	[20]
** *sseI* **	F: GATACCCCCCCTGAAATGAGTT R: GTGACAAATCGTCCAGATGCA	SPI2-intracellular replication	[20]
** *sifA* **	F: TACCACCACCGCATACCCA R: ACGAGGAACGCCTGAAACG	*Salmonella*-inducing filaments (SPI2)	[20]
** *SifB* **	F: TGATACTCAGCCTGCCCAC R: GCTCAGGGAACAAGCAAC	*Salmonella*-inducing filaments (SPI2)	[20]
** *sscA* **	F: GGCTCGCTGCGTATGTTGTT R: GCCGGCGAATTCTTTTACCT	SPI2 chaperon intracellular replication	[20]
** *qseC* **	F: GGTACCAAATTGACGCAACGTCTCAG R: GAATTCGCCCAACTTACTACGGCCTC	Sensor to adrenergic hormones	[30]
** *qseE* **	F: GGTACCAGCGACACGTTGAAGCGC R: GAATTCGCGTGTTTGTCAGATGCAGG	Sensor to adrenergic hormones	[30]
** *sdiA* **	F: AAT ATC GCT TCG TAC CAC R: GTA GGT AAA CGA GGA GCA G	Adhesion	[21]

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
