# Peer review of "Terazosin Interferes with Quorum Sensing and Type Three Secretion System and Diminishes the Bacterial Espionage to Mitigate the Salmonella Typhimurium Pathogenesis"

_antibiotics, 2022, doi:10.3390/antibiotics11040465_

Round 1

Reviewer 1 Report

Hegazy et al. examined the role of terazosin, an alpha-adrenergic antagonist, in inhibiting virulence of Salmonella enterica Typhimurium. There are several points that need to be addressed to improve the manuscript.

Major comments:

  1. The major issue with the manuscript is that the authors have only used negative control NE while they didn't use any positive control. They just used an alpha-adrenergic inhibitor terazosin and it is not clear why they selected terazosin among others apart from mentioning that in a previous study (ref 33), terazosin inhibited P. aeruginosa pathogenesis. I recommend the authors explain their selection of terazosin in more detail.
  2. The authors have determined the MIC of terazosin, however, they didn't for NE. Is there a particular reason why they didn't determine the MIC of NE? I recommend adding the data of NE MIC. How was the concentration of NE to be used determined? The authors have cited reference 56 to select 50 µg/ml NE, which, however, is not sufficient and an explanation is required. Moreover, it is not mentioned whether sub-MIC of NE was used. Apparently, the used concentration has no effect on the growth of bacteria, however; it depends upon when the growth was measured (see comment below).
  3.  It is not mentioned at what time point the growth of bacteria was measured in the presence and absence of drugs. This is critical because, with the treatment of some compounds, an initial decrease in growth is observed, which becomes indistinguishable as time passes. I would recommend showing the growth curve over time which is more convincing to show that the used concentrations of the drugs do not affect the growth of the bacteria. (same comment for Figure 5A.)
  4. Figure 2A does not say that terazosin decreased invasion. If we take a look at the untreated cell, we can see the right cell is not stained. Terazosin-treated cells are all stained compared to the untreated. Therefore, please modify the conclusion drawn from this figure.
  5. Provide scale bars for all microscopic figures.

Minor comments:

  1. I recommend English language proofreading of the entire manuscript.
  2. Section 2.1 title: There are spelling mistakes, please correct them: Minimum, Inhibitory
  3. Line 119: redundant phrase 'biofilm formation', please modify.
  4.  Line 121: change 'reduced significantly' to significantly reduced'.
  5. Line 329-330: What is the meaning of this sentence?

Author Response

Dear Reviewer,

We appreciate the reviewer for valuable constructive comments and suggestions, which greatly helped us improve the manuscript.

Best Regards

Reviewer 2 Report

The manuscript “Terazosin interferes with quorum sensing and secretion system, type three and diminishes the bacterial espionage to mitigate the Salmonella Typhimurium pathogenesis” by Hegazy et. al. is suitable for publication in Antibiotics after addressing the following minor comments: 1. Authors should provide data including another adrenergic antagonist as a control. 2. Authors should perform another in-vivo Evaluation of Terazosin Effect on the S. Typhimurium Pathogenesis for studying dose response to figure out Terazosin dose needed for 100% survival.

Author Response

(The authors gave the same response as above.)

Reviewer 3 Report

Please check and revise the scientific name of bacteria and symbol, there are founded some miss-writing.

Author Response

(The authors gave the same response as above.)

Reviewer 4 Report

The major concern is the similarity with a previous paper published also in Antibiotics (2022, 11(2), 178; https://doi.org/10.3390/antibiotics11020178). The general approach is quite the same – the authors changed in stead the target microorganisms.

Line 102 – The authors are asked to emphasize better the novelty of their work in the Introduction section

Line 104 – Please correct the title

Line 110 – To assess the effect of sub-MIC concentration on bacterial cells a viability test is necessary and not a turbidity measurement (dead cells or live cells cannot be discriminated using a spectrophotometric assay). So, the authors are asked to perform a viability assay just to be sure that ½ MIC is the right sub-MIC concentration. From this point further, the rest of the experiments are uncertain if the effect of terazosin on bacterial cells is not bullet proof assessed.

Line 130 – please mention in the figure capture for A which was the time for the growth assessment.

Line 199 – Why C violaceum was used in this paper in regard with the title? Are the same results previously presented in Antibiotics, 2022, 11(2), 178; https://doi.org/10.3390/antibiotics11020178)? The same for molecular docking experiments? Please explain.

Line 415 – Why Chromobacterium violaceum was used in this work? Is not a repetition from a previous article already published?

Line 417 – The authors are asked to give more detail about the terazosin concentrations used in the experiments. Also, for MIC determination a more sensitive method should be used (e.g. broth microdilution method).

Line 430 – please explain in the manuscript why norepinephrine was tested?

Line 435 – please mention when the samples were taken for OD measurements.

Line 444 – it is not clear – the mentioned concentrations (NE - 50 μg/mL and terazosin sub-MIC) were final concentrations in 200 μL or in the final volume (300 μL)?

Line 539 – the significance of the tested genes should be provided in the table for a better understanding of the manuscript.

Line 547 – one active site - CviR C.violaceum (PDB: 3QP5) – was not presented in a previous paper (Repurposing α-Adrenoreceptor Blockers as Promising Anti-Virulence Agents in Gram-Negative Bacteria) published also in Antibiotics?

Author Response

(The authors gave the same response as above.)

Reviewer 5 Report

Comments on the manuscript „Terazosin interferes with quorum sensing and secretion system, …..”

Terazosin is an alpha-adrenoreceptor blocker. It is known that this class of substances are not only active in the human body but also in bacteria like Salmonella. The authors carefully study the effect of terazosin on the virulence of Salmonella. The sub-MIC concentration used (2mg/ml) is several fold higher than the 2 to 20 mg daily dose used in medicine for a patient. In contrast to norepinephrine terazosine inhibited adhesion, biofilm formation and invasion of HeLa cells. Also replication inside macrophages was diminished. In mice a reduced pathogenicity was observed with terazosin. In Chromobacterium violaceum production of violacein was lowered by the drug which is expected from a quorum sensing inhibitor. In line with these results, in Salmonella sdiA, qscE and qseE expression was inhibited while norepinephrine had a stimulating effect. Also a clear inhibiting effect was observed on both T3SS secretion systems.

It is acknowledged that in the conclusions the necessity of further research is mentioned -  but it should be mentioned that relatively high concentrations of terazosin are necessary to obtain the observed effects.

Minor Comments

Fig.1A, I would have preferred to see a growth curve under the influence of terazosin. Looking into the methods part the growth experiment is described at length with all statistics applied and references to previous work, but incubation time and aeration is not mentioned.

Fig.3 B, determination of intracellular replication - the description of the method was difficult to understand for me. Possibly rephrase.

Fig. 4, The “Control (uninfected)” means “Control terazosin, uninfected”? Seeing that major differences were observed on the fifth day it would be necessary to see if terazosin has a real long lasting effect. For this, survival should be observed for some days more until the values are stable. How did the mice overcome the treatment with such high doses of terazosin?

Line 293, please indicate which genes belong to T3SS1 and which genes belong to T3SS2

Line 328, delete in the end “is”

Line 329/30,  rephrase

Line 335, norepinephrine meant

Line 380 -382 sentence too complicated  - rephrase

Line 383/384  “hire” use a different wording

Line 424/425 rephrase

Line 446: with shaking?

Line 525: At which time point in the growth curve the cells were harvested for RNA preparation?

Line 533, The primers used in this study was are listed …

Author Response

(The authors gave the same response as above.)

Round 2

Reviewer 1 Report

The reviewer is still not convinced by figure 3A of the revised manuscript. If it is the representative picture, can the authors provide more clear and easy to conclude type figure?

Author Response

Dear Reviewer,

We are very thankful for your comment, please find the attached reply.

Best Regards,

Wael

Reviewer 4 Report

The authors failed to emphasize the novelty of their work in the Introduction section.

There is no Figure 5B to show viable counting of terazsoisn-treated, NE-treated or untreated control S. Typhomurium cultures.

Line 119 – please mention in the figure capture for A which was the time for the growth assessment. Why for OD assessment was used a 24 h time and for CFU counting 16h?

I have no doubt that C. violaceum CV026 is routinely employed as a biosensor to assess the QS due to its ability to release the pigment violacein in response to acyl-homoserine lactones under the CVi/R QS system control. My question is why those data were introduced in the manuscript in regard with the paper title - Terazosin interferes with quorum sensing and secretion system, type three and diminishes the bacterial espionage to mitigate the Salmonella Typhimurium pathogenesis.

Also, the authors the authors’ state in the response letter: In the current work, the effects of terazosin and NE were evaluated on C. violaceum pigment production (which was not performed in the previous study). However, in the previous article (https://mdpi-res.com/d_attachment/antibiotics/antibiotics-11-00178/article_deploy/antibiotics-11-00178.pdf) at page 16 we may find the next subtitle - 2.4. Terazosin Inhibited the Violacein Production (The violacein production was evaluated in the absence or presence of terazosin at sub-MIC.) – so is very confusing for me.

Author Response

Dear Reviewer,

We are very thankful for your constructive comments. Please find the attached reply.

Best Regards,

Wael

Round 3

Reviewer 4 Report

Line 106 – please remove the last phrase from the manuscript – it is redundant.